# NETWORK OF PATTERNS: TIME SERIES FORECASTING WITH PATTERNS PASSING

## ABSTRACT

Time series contain diverse pattern information, and many studies have leveraged these patterns to enhance representations for more accurate forecasting. A key challenge lies in how to organize multi-scale patterns for effective information aggregation. Previous studies typically partition sequences into multi-scale pattern segments and organize them into chain or tree structures, employing neural networks to aggregate features and improve predictive performance. However, information transmission in chain structures is strictly linear and accumulative, while tree structures can aggregate multiple patterns but remain constrained by hierarchical limitations. Moreover, segments at the same or neighboring scales do not necessarily exhibit strong dependencies.

To overcome these limitations, we propose the Network of Patterns (NoP), which flexibly connects all relevant pattern segments to enable interactions between any nodes. We further introduce a Pattern Passing strategy to efficiently propagate and aggregate pattern information across this network, achieving more comprehensive integration. Experimental results demonstrate that NoP not only effectively encapsulates informative pattern signals but also establishes new state-of-the-art performance on multiple time series forecasting benchmarks, surpassing chain- and tree-based methods.

## 1 INTRODUCTION

Recently, time series forecasting has been widely applied in fields such as weather forecasting (Allen et al., 2025), energy dispatch (Hu et al., 2022), traffic flow (Jiang et al., 2021), and other areas (Luo et al., 2021; Cheng et al., 2022). Time series contain diverse pattern information, and many studies have leveraged these patterns to enhance representations for more accurate forecasting. A key challenge lies in how to organize multi-scale patterns for effective information aggregation.

As shown in Figure 1, previous studies typically partition sequences into multiscale pattern segments and organize them into chains (Wu et al., 2023; Wang et al., 2024a) or tree structures (Wu et al., 2024) to aggregate information within the same scale or between adjacent scales, thus enriching feature representations and improving forecast accuracy.

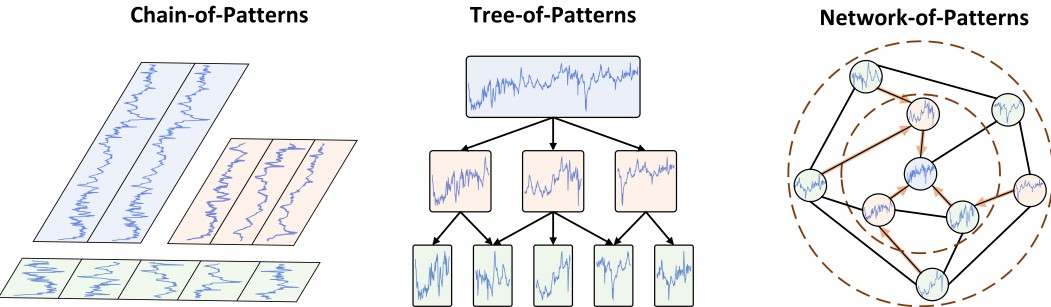

Figure 1: Three types of pattern-segment structures: Chain-of-Patterns (**left**), Tree-of-Patterns (**middle**) and Network-of-Patterns (**right**).

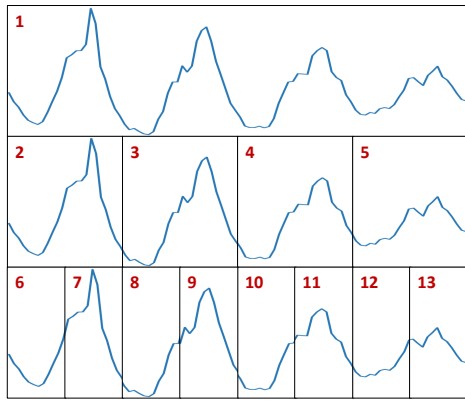

(a) Pattern segments from the Electricity dataset.

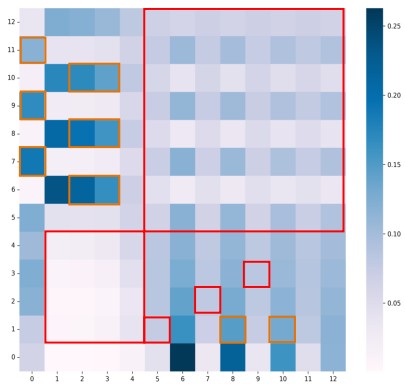

(b) Attention scores between segments

Figure 2: We design a model called SegmentTST (Segment Time Series Transformer; see Appendix A for details). The model applies Fourier analysis to identify the top-$K$ dominant periods and partitions the sequence into pattern segments (Figure 2a). These segments are then zero-padded, aligned, and fed into Transformer to model inter-segment dependencies. By visualizing the attention scores of SegmentTST trained on the Electricity dataset (UCI)(Figure 2b), we can observe that segments of the same or neighboring scales do not necessarily exhibit strong dependencies (red box), whereas cross-scale dependencies may exist even when segments neither overlap nor are nested (orange box).

However, information transmission in chain structures is strictly linear and accumulative while tree structures can aggregate multiple patterns but remain constrained by hierarchical limitations. Moreover, segments at the same scale or neighboring scales do not necessarily exhibit strong dependencies. In contrast, cross-scale dependencies may exist even when segments neither overlap nor are nested (See Figure 2a).

To address these limitations, we introduce the Network of Pattern (NoP). First, we partition the time series into multi-scale segments and construct the pattern network, which flexibly connects all relevant pattern segments via edges defined by Spectrum KL Divergence (SKL), a divergence measuring their discrepancy in the frequency domain. Each edge of the network links pattern with potential dependencies. To aggregate information across the entire network, we incorporate a learnable virtual pattern embedding that is pointed to by all segments. Subsequently, information across patterns is fused through stacked pattern passing blocks. Our experiments demonstrate that NoP not only markedly encapsulates informative pattern information, but also achieves new state-of-the-art (SOTA) results on multiple time series forecasting benchmarks (Zhou et al., 2021; Spyros Makridakis, 2018) compared with chain-and tree-based methods.

**Key Contributions:**

- We organize multi-scale patterns into a network structure and to construct edges by using Spectrum KL Divergence, thereby overcoming the limitations of chain and tree-based structures.
- We propose the Pattern Passing mechanism, which leverages network-based message passing to enhance pattern representation and aggregation. Experiments demonstrate, both qualitatively and quantitatively, that this mechanism effectively strengthens pattern modeling.

## 2 PRELIMINARIES AND RELATED WORK

### 2.1 PROBLEM DEFINITION

Let $\mathbf{X} = [\mathbf{x}_1, \mathbf{x}_2, \ldots, \mathbf{x}_H]^\top \in \mathbb{R}^{H \times C}$ be a multivariate time series, where $\mathbf{x}_t \in \mathbb{R}^C$ is the observation of $C$ variables at time step $t$. The forecasting task seeks to predict the next $L$ steps:

$\mathbf{Y} = [\mathbf{x}_{H+1}, \mathbf{x}_{H+2}, \ldots, \mathbf{x}_{H+L}]^\top \in \mathbb{R}^{L \times C}$. We train a model to output $\hat{\mathbf{Y}}$ that approximates the ground truth $\mathbf{Y}$.

## 2.2 GRAPH NEURAL NETWORKS

A graph is defined as $G = (V, E)$, with node set $V = \{v_1, v_2, \ldots, v_N\}$ and edge set $E$. Graph Neural Networks (GNNs) learn node embeddings by iteratively aggregating features from local neighborhoods via a message-passing scheme.

Let $\mathbf{h}_i^{(0)} = \mathbf{x}_i$ be the initial feature of node $v_i$. At layer $l$, the node embedding $\mathbf{h}_i^{(l)} \in \mathbb{R}^d$ is updated as

$$\mathbf{h}_i^{(l)} = \text{UPDATE}^{(l)}\left(\mathbf{h}_i^{(l-1)}, \text{AGGREGATE}^{(l)}\left(\{\mathbf{h}_j^{(l-1)} \mid j \in \mathcal{N}(i)\}\right)\right), \tag{1}$$

where $\mathcal{N}(i)$ denotes the neighbors of node $v_i$. The function $\text{AGGREGATE}^{(l)}(\cdot)$ applies a permutation-invariant operation (e.g. mean, sum, max) to collect messages from neighbors, and $\text{UPDATE}^{(l)}(\cdot)$ combines these messages with the previous state of the node to produce the new embedding.

## 2.3 RELATED WORK

**Pattern Segment based Models.** Recent years have witnessed numerous works that segment time series into subsequences and organize them into chain or tree structures for feature extraction. Timesnet (Wu et al., 2023), PDF (Dai et al., 2024), AMD (Hu et al., 2025a) and TimeMixer++ (Wang et al., 2024a) decompose time series into multi-scale pattern segments and concatenate them with identical pattern scales to capture 2D temporal variations in series exhibiting multi-period features. Peri-midFormer (Wu et al., 2024) arranges multi-scale periodic segments into pyramid structures, utilizing periodic pyramid attention mechanisms to reveal latent relationships between periodic segments at different scales. In contrast to existing pattern segmentation and feature propagation approaches, this paper introduces a novel network-structured temporal representation that enables flexible cross-pattern interactions through network message passing, achieving significant improvements in long- and short-term forecasting performance.

**Graph Neural Networks based Models.** GNNs have achieved notable success in time series forecasting (Cini et al., 2025) by leveraging message-passing to infer inter-channel relationships from multivariate time series based on temporal order (Wu et al., 2021). STGCN (Yu et al., 2017), DCRNN (Li et al., 2018) and MTGNN (Wu et al., 2020) integrate time series to structural graph to capture spatiotemporal dependencies. StemGNN (Cao et al., 2020) and FourierGNN (Yi et al., 2023) further combine graph convolution with Fourier transform to model both structural and dynamic features, capturing complex channel-wise interactions. In contrast to earlier point-based graph constructions, MSGNet (Cai et al., 2024) extracts multi-scale temporal features using segment-based modeling, improving temporal pattern recognition. Ada-MSHyper (Shang et al., 2024) downsamples the sequence into multiple scales and builds a hypergraph at each scale based on single time steps, enabling both intra- and inter-scale interactions. TimeFilter (Hu et al., 2025b) constructs a graph over all patches across channels, removes irrelevant correlations through filtering, and leverages a graph neural network to model spatio-temporal dependencies.

However, MSGNet only considers intra-scale interactions while neglecting cross-scale dependencies. Although Ada-MSHyper incorporates inter-scale interactions, its downsampling strategy fails to capture fine-grained temporal variations. TimeFilter, while effective at modeling spatio-temporal dependencies among all patches, overlooks the multi-scale characteristics of temporal dynamics. In contrast, NoP partitions the sequence into multi-scale segments and organizes them into a network structure, thereby modeling relationships across arbitrary scales while preserving temporal variation.

## 3 METHODOLOGY

In this section, we describe three core components of our approach: (1) a divergence measuring pattern segments discrepancy in frequency-domain; (2) construction of *Pattern Network*; and (3) *Pattern Passing* mechanism for feature extraction. The overall architecture of NoP is shown in Figure 3.

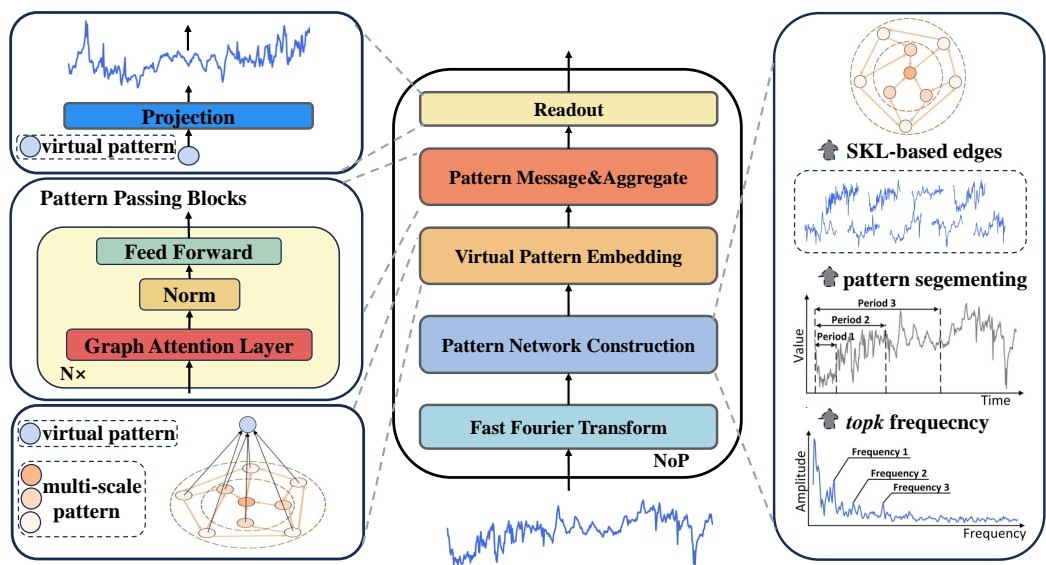

Figure 3: **The overall architeture of NoP**. The raw time series is first transformed using FFT to identify the top-$k$ frequency components, which are then used to partition the series into pattern segments. A pattern network is subsequently constructed based on SKL computed between these pattern segments. Before feeding the network into the Pattern Passing Blocks, a *virtual pattern* is inserted to aggregate information from the entire graph. After $N$-hop pattern passing, the representation of the *virtual pattern* is passed through a linear layer to produce the final output.

## 3.1 MEASURING PATTERN DISCREPANCY

Pattern segments are of variable lengths; thus, methods such as cosine similarity or Pearson correlation require padding for length alignment, which may introduce bias into the results. Fortunately, prior work (Yang & Wei, 2011; Fang et al., 2012) has demonstrated that zero-padding can align spectra without distorting their shapes, allowing us to compare pattern similarity directly in the frequency domain.

To this end, we propose Spectrum KL divergence, a divergence to quantify pattern similarity in the frequency domain. Given $N$ pattern segments of variable lengths, we first zero-pad them to obtain aligned segments$\{P_i\}_{i=1}^N$ with uniform length $H$. Unlike previous approaches that rely on pointwise differences (Wang et al., 2025b;a), we normalize the spectra into probability distributions and employ Kullback–Leibler (KL) divergence for comparison:

$$\mathrm{DKL}(P_a \,\|\, P_b) = \mathrm{KL}\big(\phi\big(\mathcal{F}(P_a)\big) \,\|\, \phi\big(\mathcal{F}(P_b)\big)\big) \qquad (2)$$

where $\mathcal{F}$ and $\phi(\cdot)$ denote the Discrete Fourier Transform operator and softmax function, respectively. This enables us to measure pattern discrepancy holistically, based on the overall distribution of frequency components(Kudrat et al., 2025). We then construct a pattern discrepancy matrix $\mathbf{D} \in \mathbb{R}^{N \times N}$ for network pruning, where $\mathbf{D}_{ab} = \mathrm{DKL}(P_a \,\|\, P_b)$.

## 3.2 PATTERN NETWORK CONSTRUCTION

We decompose the time series into seasonal and trend components, and construct the pattern network on the seasonal component where patterns are more clearly expressed. For each channel, given its seasonal component $\mathbf{X}_{season} \in \mathbb{R}^H$, we compute its amplitude spectrum:

$$A = \mathrm{Amp}\big(\mathrm{FFT}(X_{\text{season}})\big), \quad \{f_1, \ldots, f_k\} = \arg \mathrm{Topk}(A), \quad p_i = \lceil \frac{H}{f_i} \rceil,$$

where $\mathrm{FFT}(\cdot)$ is the Fast Fourier Transform (FFT) operator and $\mathrm{Amp}(\cdot)$ is the amplitude calculation function against the frequency spectrum. We enforce the frequency component $f_k = 1$ so that

periods cover all scales and then obtain $k$ period lengths $\{p_1, \ldots, p_k\}$. Each frequency $f_i$ yields $f_i$ segments $\{S_i^n\}_{n=1}^{f_i}$ of length $p_i$, which are zero-padded to length $H$ to produce aligned nodes $\{P_i, \ldots, P_N\}$, where $N = \sum_i f_i$.

Using the discrepancy $\mathbf{D} = [d_{a,b}]_{N \times N}$, We prune the pattern network by retaining the $k$ edges with the smallest divergences:

$$\mathbf{E} = \left\{ (b \to a) \mid d_{a,b} \in \text{Bottomk}\big(\{d_{u,v} \mid u \neq v\}\big) \right\}, \quad k = \lceil m(N-1)N \rceil, \tag{3}$$

where $d_{a,b}$ denotes the discrepancy between $P_a$ and $P_b$. Here $k = \lceil m(N-1)N \rceil$ for sampling ratio $m \in (0, 1]$, yielding a sparse directed network. This edge construction encourages node to aggregate pattern from highly relevant neighbors in the pattern network.

### 3.3 Pattern Passing

**Virtual Pattern Embedding.**  To globally aggregate patterns from all pattern segments, we insert a *virtual pattern* $P_0$ into the pattern graph. We denote the *virtual pattern* embedding after the $l$-th Pattern Passing Block as $h_V^{(l)} \in \mathbb{R}^D$. This learnable embedding continuously aggregates information from all pattern segments during the pattern passing process.

**Pattern Passing Block.**  We then stack $L$ Pattern Passing Blocks. Denote $h_i^{(0)} \in \mathbb{R}^D$ the embedded input of node $i$. Each block updates

$$h_i^{(l)} = \text{MLP}\Big(\text{LN}\big(\text{GAT}\big(h_i^{(l-1)}, \{h_j^{(l-1)} \mid j \to i\}\big)\big)\Big), \tag{4}$$

where graph-attention layer (Velickovic et al., 2017) (GAT) attends to neighbors $(j \to i)$, followed by layer norm (LN) and an MLP.

**Readout.**  After $L$ blocks, we extract the *virtual pattern* embedding $h_V^{(L)}$ and project to produce the seasonal forecast: $\hat{Y}_{\text{season}} = \text{Linear}\big(h_V^{(L)}\big)$. We similarly project the trend component and sum to obtain the final output: $\hat{Y} = \hat{Y}_{\text{season}} + \hat{Y}_{\text{trend}}$.

## 4 Experiments

We conducted extensive experiments to investigate the following research questions: **RQ1**: Does organizing pattern segements into a network structure outperform chain and tree structures? **RQ2**: Does the Patterns Passing mechanism in the NoP enhance the patterns and lead to improved predictive performance? **RQ3**: Can NoP achieve performance comparable to or even surpass that of representative forecasting models?

### 4.1 Experimental Settings

**Datasets.**  For long term forecasting, we adopt 7 real-world benchmark datasets from various application scenarios, including ETT (Zhou et al., 2021),Weather (Wetterstation), ECL (UCI) and Exchange (Lai et al., 2018). The look-back window $H$ is fixed at 96, and the forecasting horizon $L$ is set to $\{96, 192, 336, 720\}$ for all long-term forecasting tasks. For short term forecasting, we adopt M4 benchmark (Spyros Makridakis, 2018) which contains the yearly, quarterly and monthly collected univariate marketing data.

**Baselines.**  To address RQ3, we compare NoP with two categories of baselines. The first include **pattern segment-based models** such as TimesNet (Wu et al., 2023), PDF (Dai et al., 2024), and Peri-midFormer (Wu et al., 2024). The second covers **representative forecasting models**, including GNN-based approaches (TimeFilter, MSGNet (Hu et al., 2025b; Cai et al., 2024)), Transformer-based methods (TQNet, iTransformer, PatchTST (Lin et al., 2025; Liu et al., 2023; Nie et al., 2022)), and an MLP-based method (DLinear (Zeng et al., 2023)). Notably, both TimesNet and PDF adopt CNN architectures as part of their design. Some **pattern segement-based** methods were not included because their architectural designs make it difficult to isolate the effect of the chain (or tree)

structure on predictive performance. All the experiments are conducted three times to eliminate randomness. More details about the datasets and evaluation metrics we used are listed in Appendix B.1 and Appendix B.2, respectively. In Appendix B.3, we detail the network structure and training hyperparameter configuration used in the experiment.

## 4.2 NETWORK V.S. CHAIN AND TREE (RQ1)

**Setup.** We evaluated the impact of different pattern segments structures on long-term forecasting task. To address **RQ1**, we controlled the attention masks of SegmentTST(More details in Appendix A) to organize the pattern segments into chain, tree, and network structures, denoted as SegmentTST(*Chain*), SegmentTST(*Tree*), and SegmentTST(*Network*), respectively. Except for the different attention mask used during the attention, these models share the same model settings.

Table 1: Long term forecasting task on different **pattern segement structures**. The results are avaergaed from four different forecasting length {96, 192, 336, 720}. Red: best, Blue: second best. The full results are shown in Table 7.

| Datasets | ETTm1 | | ETTm2 | | ETTh1 | | ETTh2 | | Weather | | Electricity | | Exchange | |
| Methods | MSE | MAE | MSE | MAE | MSE | MAE | MSE | MAE | MSE | MAE | MSE | MAE | MSE | MAE |
| --- | --- | --- | --- | --- | --- | --- | --- | --- | --- | --- | --- | --- | --- | --- |
| SegmentTST(*Chain*) | 0.405 | 0.413 | 0.338 | 0.386 | 0.469 | 0.465 | 0.497 | 0.485 | 0.247 | 0.300 | 0.199 | 0.292 | **0.339** | 0.398 |
| SegmentTST(*Tree*) | 0.403 | 0.413 | 0.344 | 0.389 | 0.461 | 0.462 | 0.491 | 0.481 | 0.258 | 0.297 | 0.191 | 0.283 | 0.385 | 0.415 |
| SegmentTST(*Network*) | 0.399 | 0.413 | 0.327 | 0.376 | 0.455 | 0.456 | 0.467 | 0.467 | 0.244 | 0.297 | 0.185 | 0.278 | 0.353 | 0.406 |
| **NoP** | **0.377** | **0.395** | **0.278** | **0.324** | **0.434** | **0.439** | **0.373** | **0.402** | **0.244** | **0.274** | **0.167** | **0.260** | 0.349 | **0.396** |

**Results** As shown in Table 1, the SegmentTST(*Network*) consistently achieves superior performance in long-term forecasting, with noticeable improvements over both SegmentTST(*Chain*) and SegmentTST(*Tree*) across all seven datasets. This indicates that organizing pattern segments into a network structure is generally more suitable for forecasting tasks. As an enhanced version of SegmentTST(*Network*), NoP further improves predictive performance by leveraging the Pattern Passing mechanism to better capture and aggregate global information within the network.

## 4.3 NOP V.S. BASELINES.

**Setup** We conducted both long- and short-term forecasting tasks on pattern segment-based models and representative forecasting models to comprehensively assess their predictive performance. Furthermore, we decomposed the forecasting results to assess how effectively NoP enhances pattern representations and aggregations.

**Visualization (RQ2).** We decompose the forecasting results of TimesNet, PDF, Peri-midFormer, and NoP on the Electricity and ETTh2 datasets into trend and seasonal components, and visualize the seasonal part, as the pattern network is constructed solely on the seasonal component. The visualization is shown in Figure 4. As observed, NoP produces seasonal components that are closer to ground truth, highlighting the effectiveness of combining the pattern network with the pattern passing mechanism. In addition, NoP achieves predictive performance that matches or even exceeds that of other representative forecasting models on multiple benchmarks.

### 4.3.1 RESULTS (RQ3)

Tables 2 and 3 show that NoP achieves state-of-the-art performance among **pattern-segment-based** models that based on chain or tree structures, particularly on the ETTm1, ETTh2, and Electricity datasets. Unlike ETTh2 and Electricity, which exhibit relatively monotonic and well-defined patterns, ETTm1 contains more diverse and complex dynamics, making it especially challenging to accurately capture inter-segment dependencies. By flexibly connecting correlated pattern segments, NoP overcomes the inherent hierarchical limitations of chain- or tree-based structures and delivers significantly stronger performance on ETTm1 than on ETTh2 and Electricity. Moreover, NoP consistently outperforms representative forecasting baselines across most datasets, highlighting its superior generalization ability.

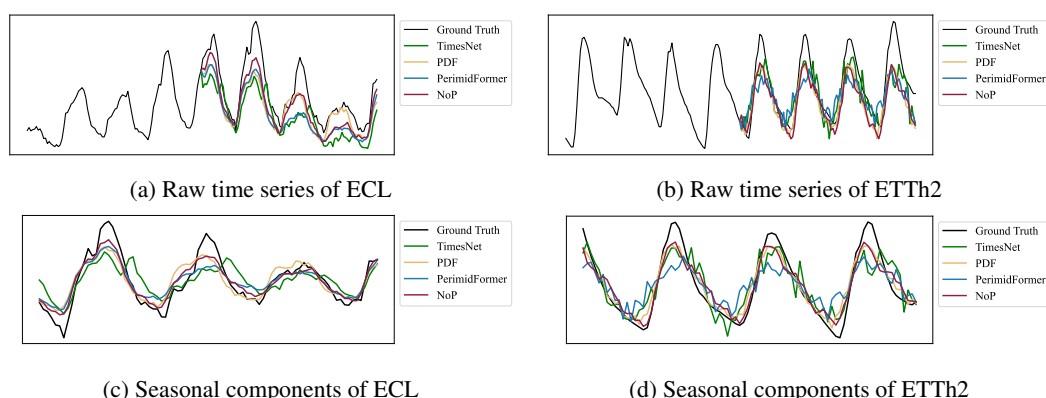

(a) Raw time series of ECL        (b) Raw time series of ETTh2

(c) Seasonal components of ECL      (d) Seasonal components of ETTh2

Figure 4: The visualization of raw time series and decomposed seasonal components from raw time series, compared with TimesNet, PDF and Peri-midFormer.

Table 2: Long term forecasting task on **pattern segement-based models** and **representative forecasting models**. The results are avergaed from four different forecasting length {96, 192, 336, 720}. Red: best, Blue: second best. (* means former). The full results are shown in Table 8.

| Methods | NoP (**Ours**) | TimesNet (2023) | PDF (2024) | Peri-mid* (2024) | TimeFilter (2025b) | MSGNet (2024) | TQNet (2025) | iTrans* (2023) | PatchTST (2022) | DLinear (2023) |
|---|---|---|---|---|---|---|---|---|---|---|
| Metric | MSE MAE | MSE MAE | MSE MAE | MSE MAE | MSE MAE | MSE MAE | MSE MAE | MSE MAE | MSE MAE | MSE MAE |
| ETTm1 | **0.377 0.395** | 0.412 0.416 | 0.395 0.403 | 0.409 0.410 | 0.381 0.396 | 0.398 0.411 | 0.390 0.401 | 0.407 0.412 | 0.388 0.402 | 0.403 0.407 |
| ETTm2 | 0.278 0.324 | 0.295 0.333 | 0.287 0.331 | 0.290 0.328 | **0.275 0.323** | 0.288 0.330 | 0.281 0.324 | 0.292 0.336 | 0.291 0.335 | 0.346 0.396 |
| ETTh1 | **0.434** 0.439 | 0.476 0.465 | 0.435 **0.433** | 0.455 0.446 | 0.464 0.446 | 0.452 0.452 | 0.448 0.437 | 0.462 0.454 | 0.451 0.449 | 0.461 0.458 |
| ETTh2 | **0.373 0.402** | 0.417 0.428 | 0.376 0.402 | 0.400 0.416 | 0.393 0.412 | 0.396 0.417 | 0.387 0.406 | 0.383 0.407 | 0.387 0.412 | 0.559 0.518 |
| Weather | 0.244 0.274 | 0.262 0.288 | 0.254 0.277 | 0.262 0.283 | **0.243 0.272** | 0.249 0.278 | 0.249 0.275 | 0.260 0.281 | 0.259 0.281 | 0.266 0.318 |
| ECL | **0.167 0.260** | 0.196 0.296 | 0.218 0.300 | 0.178 0.267 | 0.167 0.263 | 0.194 0.300 | 0.177 0.270 | 0.175 0.267 | 0.204 0.294 | 0.225 0.319 |
| Exchange | **0.349 0.396** | 0.418 0.443 | 0.353 0.398 | 0.388 0.417 | 0.376 0.410 | 0.399 0.430 | 0.363 0.404 | 0.377 0.415 | 0.373 0.409 | 0.371 0.423 |
| Average | **0.317 0.356** | 0.354 0.381 | 0.331 0.363 | 0.340 0.367 | 0.328 0.360 | 0.339 0.374 | 0.328 0.360 | 0.337 0.367 | 0.336 0.369 | 0.376 0.405 |

Table 3: Short term forecasting task on **pattern segement-based models** and **representative forecasting models**. The prediction lenghs are {6, 48} and results are weighted averaged from several datasets under different sample intervals. The full results are shown in Table 9. (TMixer is TimeMixer)

| Methods | NoP (**Ours**) | TimesNet (2023) | Peri-mid* (2024) | iTrans* (2023) | PatchTST (2022) | DLinear (2023) | LightTS (2022) | Pyra* (2022a) | Stationary (2022b) | FED* (2022) | TMixer (2024b) | Re* (2020) |
|---|---|---|---|---|---|---|---|---|---|---|---|---|
| SMAPE | **11.837** | 11.888 | 11.897 | 13.233 | 12.866 | 12.500 | 11.962 | 13.616 | 12.780 | 12.605 | 11.885 | 12.805 |
| MASE | **1.596** | 1.607 | 1.607 | 1.850 | 1.734 | 1.678 | 1.609 | 1.843 | 1.756 | 1.677 | 1.598 | 1.777 |
| OWA | **0.854** | 0.858 | 0.859 | 0.972 | 0.928 | 0.899 | 0.862 | 0.984 | 0.930 | 0.903 | 0.856 | 0.937 |

## 4.4 ABLATION STUDIES

**Setup.** To assess the contribution of key components in NoP, we performed ablation studies on long-term forecasting tasks. The results are presented in Table 4.The table reports the impact on performance when different components of NoP are removed. **w/o FFT** constructs the pattern network by directly segmenting the time series into patches, without applying FFT-based segmentation. **w/o Network** organizes the segmented patterns into a tree structure instead of building a network. **w/o PPB** performs attention mechanism to aggregate information. **w/o Virtual Pattern** applies average pooling over the network without virtual pattern, and uses the pooled representation for forecasting.

Table 4: Ablation studies on long-term forecasting task to assess the key components of NoP. The results are avergaed from four different forecasting length {96, 192, 336, 720}. **Red**: best, Blue: second best.

| Datasets | ETTm1 | | ETTm2 | | ETTh1 | | ETTh2 | | Electricity | | Weather | | Exchange | |
| Variants | MSE | MAE | MSE | MAE | MSE | MAE | MSE | MAE | MSE | MAE | MSE | MAE | MSE | MAE |
|---|---|---|---|---|---|---|---|---|---|---|---|---|---|---|
| w/o FFT | 0.417 | 0.418 | 0.285 | 0.332 | 0.480 | 0.482 | 0.403 | 0.421 | 0.259 | 0.286 | 0.188 | 0.275 | 0.395 | 0.416 |
| w/o Network | 0.388 | 0.401 | 0.282 | 0.327 | 0.471 | 0.457 | 0.387 | 0.409 | 0.253 | 0.281 | **0.167** | **0.260** | 0.363 | 0.404 |
| w/o PPB | 0.390 | 0.402 | 0.284 | 0.329 | 0.463 | 0.454 | 0.397 | 0.413 | 0.256 | 0.282 | 0.169 | 0.263 | 0.392 | 0.420 |
| w/o Virtual Pattern | 0.386 | 0.401 | 0.282 | 0.327 | 0.455 | 0.449 | 0.391 | 0.410 | 0.251 | 0.278 | 0.167 | 0.260 | 0.371 | 0.408 |
| **NoP** | **0.377** | **0.395** | **0.278** | **0.324** | **0.434** | **0.439** | **0.373** | **0.402** | **0.244** | **0.274** | 0.167 | 0.260 | **0.349** | **0.396** |

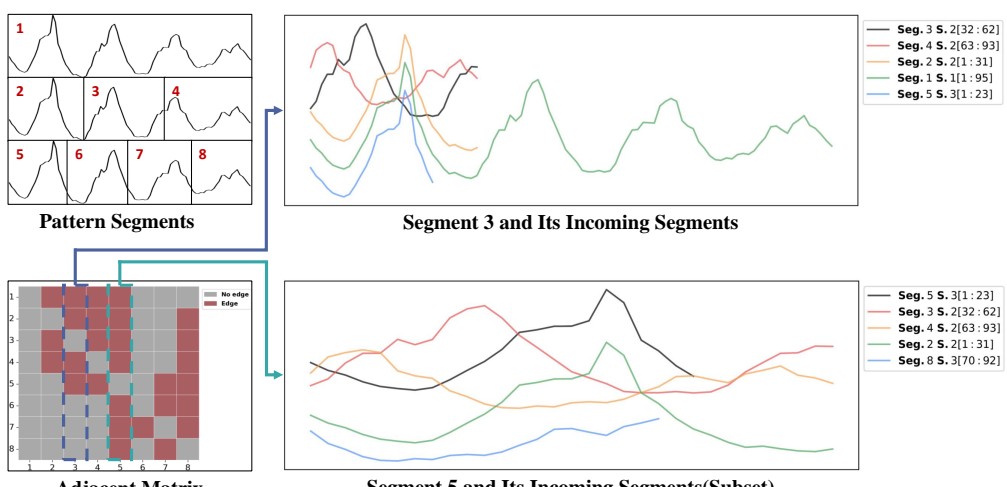

Figure 5: Visualization of two segments from the pattern network constructed on the Electricity dataset, along with their incoming connections(to better observe similarities across patterns, we vertically shifted the segments and omitted part of the incoming connections for clarity). "**Seg.** 3 **S.** 2[32:62]" denotes the 3rd **Seg**ment in the pattern network, located at the 2nd **S**cale, corresponding to time steps 32–62 in the original sequence. The pattern network flexibly links segments with fluctuations comparable to Segments 3 and 5, thereby enabling mutual reinforcement through the aggregation of similar patterns and overcoming the hierarchical constraints inherent to chain- or tree-structured models.

**Results.** The results in the Table 4 indicate that the FFT partitions pattern segments more effectively than Patch. This is because manually fixing patch lengths may inadvertently split or merge pattern segments, highlighting the need to model multi-scale segment information. In addition, the results on the Weather dataset suggest that, in a few cases, connecting pattern segments using a tree structure can also yield satisfactory performance. However, in most scenarios, the network structure enhanced with the Pattern Passing mechanism consistently achieves the best predictive performance.

## 4.5 MODEL ANALYSIS

**Case Study on Pattern Network Construction.** We analyzed the pattern network constructed on the Electricity dataset and visualized two segments along with their incoming neighbors, as illustrated in Figure 5. To better observe the similarity among different patterns, we vertically shifted the segments and omitted part of the incoming connections for clarity. The pattern network flexibly connects segments that exhibit comparable fluctuations to Segments 3 and 5, enabling them to reinforce themselves by aggregating information from similar patterns. This design overcomes the hierarchical constraints imposed by chain- or tree-structured models. We further analyze the hyper-parameters related to pattern network construction in Appendix C.

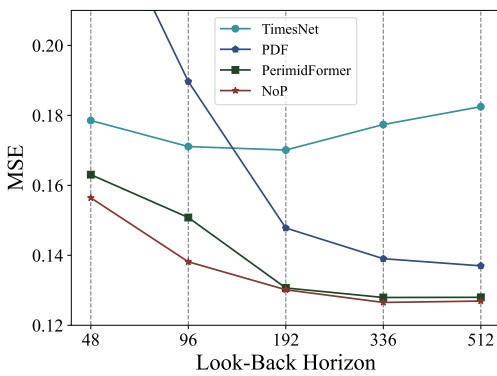 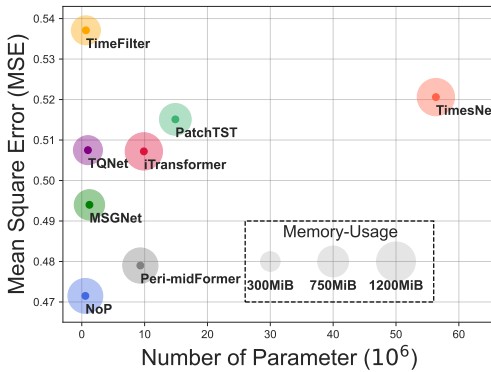

(a) Influence of look-back horizon on long term prediction task (Electricity dataset, forecasting length is 96).

(b) Complexity analysis on long term prediction task (ETTh1 dataset, forecasting length is 720).

Figure 6: Hyper-parameters sensitivity analysis (**left**) and complexity analysis (**right**).

**Influence of Look-Back Horizon.** We study the effect of input length on **pattern-segment-based models** by varying the look-back window over $\{48, 96, 192, 336, 512\}$ time steps. As shown in Figure 6a, NoP leverages the richer pattern information present in longer sequences to strengthen its pattern representations, yielding superior predictive performance compared with other models across all look-back windows. However, as the look-back window grows further, the number of novel patterns decreases and noise accumulates, which constrains all models; consequently, performance gains become marginal and can even decline.

**Efficiency Analysis.** We analyze the model complexity on the long-term forecasting task using the ETTh1 dataset with the same batch size. Specifically, we consider the number of trainable parameters, GPU memory usage, and MSE as evaluation metrics. As shown in Figure 6b, NoP achieves the lowest MSE with fewer parameters, demonstrating its potential to achieve excellent forecasting performance with less computational resources.

## 5 CONCLUSION

In this work, we introduced a method for time series forecasting called NoP. It decomposes time series into pattern segments, a metric for quantifying pattern similarity in the frequency domain, and organizes them into pattern network. The Pattern Passing mechanism is then employed to propagate and aggregate information across the network. Extensive experiments demonstrate that NoP not only markedly enhances the modeling of pattern, but also achieves new state-of-the-art results on multiple time series forecasting benchmarks compared to chain- or tree-based methods. However, like other pattern segment-based methods, NoP does not model the relationships among pattern segments across different channels, thereby overlooking the inter-variable dependencies. In future work, we plan to explore the potential for incorporating channel-wise correlations into such frameworks.

## ETHICS STATEMENT

This work adheres to the ICLR Code of Ethics. In this study, no human subjects or animal experimentation was involved. All datasets used, including ETT (Zhou et al., 2021), Weather (Wetterstation), ECL (UCI), Exchange (Lai et al., 2018) and M4 benchmark Spyros Makridakis (2018), were sourced in compliance with relevant usage guidelines, ensuring no violation of privacy. We have taken care to avoid any biases or discriminatory outcomes in our research process. No personally identifiable information was used, and no experiments were conducted that could raise privacy or security concerns. We are committed to maintaining transparency and integrity throughout the research process.

## REPRODUCIBILITY STATEMENT

We have made every effort to ensure that the results presented in this paper are reproducible. This paper describes all experimental settings in detail, including, the hyperparameters of the neural network in NoP, the model training configuration, and the dataset partitioning. We will open source the code used in this work in the future.

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

## A  ARCHITECTURE OF SEGMENTTST

We introduce SegmentTST (Segment Time Series Transformer) for time-series forecasting. The model first applies Fourier analysis to extract the top-$K$ dominant periods and partitions the original sequence into pattern segments, which are aligned via zero padding as described in Section 3.2, yielding $\mathbf{P} = \{P_i\}_{i=1}^N$. These aligned segments are then embedded before modeling inter-segment dependencies:

$$\mathbf{Z} = \mathrm{Proj}\big(\mathrm{Stack}(\mathbf{P})\big) \in \mathbb{R}^{N \times D}, \tag{5}$$

where $\mathrm{Stack}(\cdot)$ concatenates the padded pattern segments, and $\mathrm{Proj} : \mathbb{R}^H \to \mathbb{R}^D$ denotes a linear projection layer. The resulting representations are processed by a stacked Transformer encoder (Nie et al., 2022) with a masking mechanism:

$$\mathbf{Z}' = \mathrm{StackedEncoder}(\mathbf{Z}, \mathrm{Mask}), \tag{6}$$

where $\mathrm{Mask} \in \mathbb{R}^{N \times N}$ is used to block unrelated pattern segments, thereby organizing dependencies into *chain*, *tree*, or *network* structures. Specifically, the tree-structured mask follows the implementation in Peri-midFormer (Wu et al., 2024), while the network-structured mask is constructed as described in Section 3. Depending on whether a mask is applied and its type, we obtain four variants: SegmentTST, SegmentTST(*chain*), SegmentTST(*tree*), and SegmentTST(*network*).

Finally, we apply average pooling to $\mathbf{Z}'$ and pass the result through a linear layer to generate the forecast $\hat{Y}$:

$$\hat{Y} = \mathrm{Proj}\big(\mathrm{Mean}(\mathbf{Z}')\big) \in \mathbb{R}^L. \tag{7}$$

## B  IMPLEMENTATION DETAILS

In this section, we first introduce in detail the datasets used in the long-term and short-term time series forecasting tasks, including the division of training sets, test sets, and validation sets. We then further introduce the evaluation metrics used in the time series forecasting tasks. All experiments and deep neural networks training are implemented in PyTorch on 4 NVIDIA RTX 4090 24GB GPU.

### B.1  DOWNSTREAM TASKS DATASETS DETAILS

We utilize 8 datasets including ETTh1, ETTh2, ETTm1, ETTm2  (Zhou et al., 2021), Electricity (UCI), Weather  (Wetterstation), and Exchange  (Lai et al., 2018) to conduct long-term time series forecasting experiments,with a detailed description of the dataset provided in Table 5. Our model, NoP, employ input series of lookback lengths 96, with forecast horizons of {96, 192, 336, 720}. For short-term forecasting experiments, we employ the M4 (Spyros Makridakis, 2018) benchmark dataset, predicting data of various frequencies.

### B.2  METRICS

We assess the five TSA tasks using various metrics. For long-term forecasting and imputation tasks, we employ mean squared error (MSE) and mean absolute error (MAE). For short-term forecasting, we utilize symmetric mean absolute percentage error (SMAPE), mean absolute scaled Error (MASE), and overall weighted average (OWA), with OWA being a metric unique to the M4 competition. The calculations for these metrics are as follows:

$$\mathrm{MSE} = \sum_{i=1}^n \left(y_i - \hat{y}_i\right)^2, \tag{8}$$

$$\mathrm{MAE} = \sum_{i=1}^n |y_i - \hat{y}_i|, \tag{9}$$

$$\mathrm{SMAPE} = \frac{200}{T} \sum_{i=1}^T \frac{\left|\mathbf{X}_i - \hat{\mathbf{Y}}_i\right|}{|\mathbf{X}_i| + \left|\hat{\mathbf{Y}}_i\right|}, \tag{10}$$

Table 5: Dataset descriptions. The dataset size is organized in (Train, Validation, Test).

| Tasks | Dataset | Dim | Series Length | Dataset Size | Information (Frequency) |
|---|---|---|---|---|---|
| Forecasting (Long-term) | ETTm1, ETTm2 | 7 | {96, 192, 336, 720} | (34465, 11521, 11521) | Electricity (15 mins) |
| | ETTh1, ETTh2 | 7 | {96, 192, 336, 720} | (8545, 2881, 2881) | Electricity (15 mins) |
| | Electricity | 321 | {96, 192, 336, 720} | (18317, 2633, 5261) | Electricity (Hourly) |
| | Weather | 21 | {96, 192, 336, 720} | (36792, 5271, 10540) | Weather (10 mins) |
| | Exchange | 8 | {96, 192, 336, 720} | (5120, 665, 1422) | Exchange rate (Daily) |
| Forecasting (short-term) | M4-Yearly | 1 | 6 | (23000, 0, 23000) | Demographic |
| | M4-Quarterly | 1 | 8 | (24000, 0, 24000) | Finance |
| | M4-Monthly | 1 | 18 | (48000, 0, 48000) | Industry |
| | M4-Weakly | 1 | 13 | (359, 0, 359) | Macro |
| | M4-Daily | 1 | 14 | (4227, 0, 4227) | Micro |
| | M4-Hourly | 1 | 48 | (414, 0, 414) | Other |

$$\text{MAPE} = \frac{100}{T} \sum_{i=1}^{T} \frac{\left| \mathbf{X}_i - \hat{\mathbf{Y}}_i \right|}{|\mathbf{X}_i|}, \tag{11}$$

$$\text{MASE} = \frac{1}{T} \sum_{i=1}^{T} \frac{\left| \mathbf{X}_i - \hat{\mathbf{Y}}_i \right|}{\frac{1}{T-q} \sum_{j=q+1}^{T} |\mathbf{X}_j - \mathbf{X}_{j-q}|}, \tag{12}$$

$$\text{OWA} = \frac{1}{2} \left[ \frac{\text{SMAPE}}{\text{SMAPE}_{\text{Naïve2}}} + \frac{\text{MASE}}{\text{MASE}_{\text{Naïve2}}} \right], \tag{13}$$

where, $y_i$ is the ground truth value, $\hat{y}_i$ is the model prediction, $q$ is the periodicity of the time series data. $\mathbf{X}, \hat{\mathbf{Y}} \in \mathbb{R}^{T \times C}$ are the ground truth and prediction results of the future with $T$ time points and $C$ dimensions. $\mathbf{X}_i$ means the $i$-th future time point.

### B.3 EXPERIMENT CONFIGURATION OF NoP

Table 6: Experiment configuration of NoP. All the experiments use the Adam (Kingma & Ba, 2017) optimizer with the default hyperparameter configuration for $(\beta_1, \beta_2)$ as (0.9, 0.999).

| Task / Configurations | Hyper-parameters | | | Training Process | | | | |
|---|---|---|---|---|---|---|---|---|
| | k | layers | $d_{\text{model}}$ | LR | | | | Loss | Batch Size | Epochs |
| Long-term Forecasting | 2-5 | 1-3 | 64-768 | 0.001 0.0005 0.0003 0.0001 | | | | MSE | 16 | 10 |
| Short-term Forecasting | 2-5 | 1-3 | 64-768 | 0.001 0.0005 0.0003 0.0001 | | | | SMAPE | 16 | 10 |

## C HYPER-PARAMETERS SENSITIVITY ANALYSIS.

We conduct hyperparameter sensitivity analysis on the long-term forecasting task using the ETT datasets. When evaluating the sensitivity of hyperparameter $k$, we fix the edge ratio $m = 0.2$; when analyzing $m$, we fix $k = 5$. Increasing $k$ (Figure 7b) produces more diverse pattern segments, and the resulting aggregation enriches the pattern representation, thereby improving prediction performance. However, an excessive number of segments introduces noise, which interferes with pattern aggregation. A similar phenomenon is observed for the sampling ratio $m$ (Figure 7a): when $m$ is too small, important segment connections may be overlooked, while overly large values cause noisy segments to disrupt aggregation.

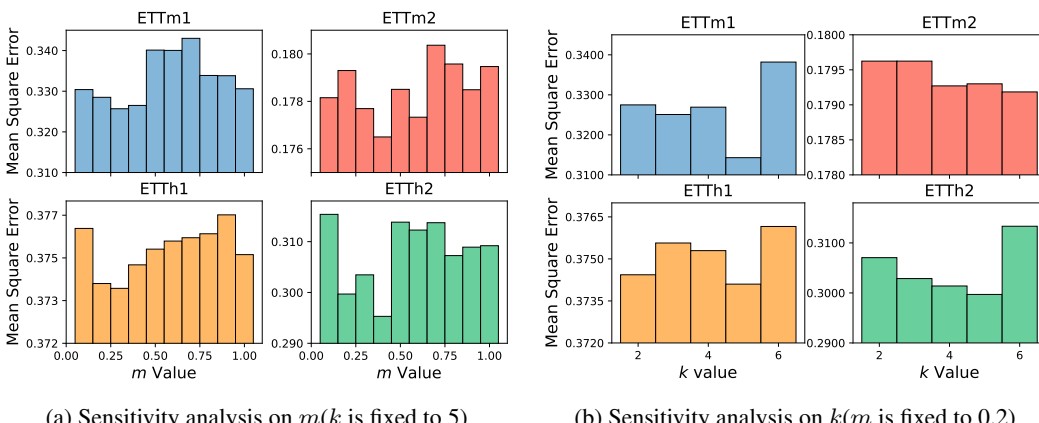

(a) Sensitivity analysis on $m$($k$ is fixed to 5)  (b) Sensitivity analysis on $k$($m$ is fixed to 0.2)

Figure 7: Hyper-parameters sensitivity analysis.

# D    FULL RESULTS

## D.1    FULL RESULTS OF DIFFERENT PATTERN SEGMENT STRUCTURES (TABLE 7)

## D.2    FULL RESULTS OF LONG TERM FORECASTING TASK (TABLE 8)

## D.3    FULL RESULTS OF SHORT TERM FORECASTING TASK (TABLE 9)

# E    LLM USAGE

Large Language Models (LLMs) were used to aid in the writing and polishing of the manuscript. Specifically, we used an LLM to assist in refining the language, improving readability, and ensuring clarity in various sections of the paper. The model helped with tasks such as sentence rephrasing, grammar checking, and enhancing the overall flow of the text.

It is important to note that the LLM was not involved in the ideation, research methodology, or experimental design. All research concepts, ideas, and analyses were developed and conducted by the authors. The contributions of the LLM were solely focused on improving the linguistic quality of the paper, with no involvement in the scientific content or data analysis.

The authors take full responsibility for the content of the manuscript, including any text generated or polished by the LLM. We have ensured that the LLM-generated text adheres to ethical guidelines and does not contribute to plagiarism or scientific misconduct.

Table 7: Full results of different pattern segment structures. The results are avergaed from four different forecasting length {96, 192, 336, 720}. **Red**: best, Blue: second best.

| Methods | | NoP | | SegmentTST(chain) | | SegmentTST(tree) | | SegmentTST(Network) | |
|---|---|---|---|---|---|---|---|---|---|
| Datasets \ Horizon | | MSE | MAE | MSE | MAE | MSE | MAE | MSE | MAE |
| ETTm1 | 96 | **0.314** | **0.354** | 0.340 | 0.376 | 0.338 | 0.376 | 0.332 | 0.375 |
| | 192 | **0.355** | **0.379** | 0.380 | 0.394 | 0.380 | 0.394 | 0.378 | 0.398 |
| | 336 | **0.392** | **0.405** | 0.413 | 0.418 | 0.411 | 0.418 | 0.409 | 0.420 |
| | 720 | **0.448** | **0.441** | 0.485 | 0.463 | 0.483 | 0.462 | 0.478 | 0.459 |
| | Avg | **0.377** | **0.395** | 0.405 | 0.413 | 0.403 | 0.412 | 0.399 | 0.413 |
| ETTm2 | 96 | **0.176** | **0.259** | 0.204 | 0.292 | 0.190 | 0.283 | 0.185 | 0.276 |
| | 192 | **0.241** | **0.302** | 0.280 | 0.352 | 0.280 | 0.357 | 0.267 | 0.337 |
| | 336 | **0.297** | **0.339** | 0.360 | 0.408 | 0.377 | 0.410 | 0.353 | 0.405 |
| | 720 | **0.397** | **0.396** | 0.507 | 0.494 | 0.530 | 0.507 | 0.505 | 0.483 |
| | Avg | **0.278** | **0.324** | 0.338 | 0.386 | 0.344 | 0.389 | 0.327 | 0.376 |
| ETTh1 | 96 | **0.374** | **0.399** | 0.389 | 0.409 | 0.382 | 0.404 | 0.375 | 0.399 |
| | 192 | **0.428** | **0.431** | 0.435 | 0.439 | 0.431 | 0.435 | 0.422 | 0.430 |
| | 336 | **0.465** | **0.457** | 0.487 | 0.472 | 0.485 | 0.471 | 0.475 | 0.465 |
| | 720 | **0.471** | **0.470** | 0.565 | 0.539 | 0.547 | 0.538 | 0.546 | 0.531 |
| | Avg | **0.435** | **0.439** | 0.469 | 0.465 | 0.461 | 0.462 | 0.455 | 0.456 |
| ETTh2 | 96 | **0.295** | **0.347** | 0.311 | 0.371 | 0.314 | 0.372 | 0.311 | 0.369 |
| | 192 | **0.367** | **0.392** | 0.416 | 0.440 | 0.418 | 0.439 | 0.416 | 0.437 |
| | 336 | **0.408** | **0.429** | 0.531 | 0.509 | 0.510 | 0.497 | 0.481 | 0.478 |
| | 720 | **0.420** | **0.438** | 0.729 | 0.618 | 0.722 | 0.615 | 0.660 | 0.584 |
| | Avg | **0.373** | **0.402** | 0.497 | 0.485 | 0.491 | 0.481 | 0.467 | 0.467 |
| Weather | 96 | **0.162** | **0.209** | 0.177 | 0.236 | 0.171 | 0.232 | 0.170 | 0.230 |
| | 192 | **0.208** | **0.251** | 0.208 | 0.270 | 0.208 | 0.273 | 0.207 | 0.271 |
| | 336 | **0.263** | **0.291** | 0.267 | 0.320 | 0.318 | 0.313 | 0.266 | 0.317 |
| | 720 | **0.343** | **0.344** | 0.334 | 0.373 | 0.335 | 0.371 | 0.333 | 0.370 |
| | Avg | **0.244** | **0.274** | 0.246 | 0.300 | 0.258 | 0.297 | 0.244 | 0.297 |
| ECL | 96 | **0.138** | **0.233** | 0.176 | 0.269 | 0.167 | 0.259 | 0.162 | 0.253 |
| | 192 | **0.155** | **0.248** | 0.184 | 0.277 | 0.175 | 0.267 | 0.170 | 0.263 |
| | 336 | **0.172** | **0.266** | 0.201 | 0.295 | 0.192 | 0.286 | 0.186 | 0.280 |
| | 720 | **0.202** | **0.294** | 0.236 | 0.328 | 0.228 | 0.321 | 0.222 | 0.314 |
| | Avg | **0.167** | **0.260** | 0.199 | 0.292 | 0.191 | 0.283 | 0.185 | 0.277 |
| Exchange | 96 | **0.081** | **0.199** | 0.081 | 0.202 | 0.082 | 0.203 | 0.080 | 0.202 |
| | 192 | **0.173** | **0.295** | 0.170 | 0.308 | 0.168 | 0.303 | 0.204 | 0.320 |
| | 336 | **0.320** | **0.407** | 0.295 | 0.412 | 0.473 | 0.471 | 0.318 | 0.419 |
| | 720 | **0.823** | **0.683** | 0.810 | 0.671 | 0.818 | 0.681 | 0.808 | 0.685 |
| | Avg | 0.349 | **0.396** | **0.339** | 0.398 | 0.385 | 0.414 | 0.353 | 0.406 |
| Average | | **0.317** | **0.356** | 0.359 | 0.390 | 0.358 | 0.387 | 0.346 | 0.381 |

Table 8: Full results of long term forecasting task on **pattern segement-based models** and **representative forecasting models**. The results are avergaed from four different forecasting length {96, 192, 336, 720}. **Red**: best, Blue: second best. (* means former)

| Methods / Datasets | | NoP (Ours) | | Timesnet (2023) | | PDF (2024) | | Peri-mid* (2024) | | TimeFilter (2025b) | | MSGNet (2024) | | TQNet (2025) | | iTrans* (2023) | | PatchTST (2022) | | DLinear (2023) | |
|---|---|---|---|---|---|---|---|---|---|---|---|---|---|---|---|---|---|---|---|---|---|
| Horizon | | MSE | MAE | MSE | MAE | MSE | MAE | MSE | MAE | MSE | MAE | MSE | MAE | MSE | MAE | MSE | MAE | MSE | MAE | MSE | MAE |
| ETTm1 | 96 | 0.314 | 0.354 | 0.331 | 0.372 | 0.336 | 0.371 | 0.334 | 0.370 | 0.321 | 0.359 | 0.319 | 0.366 | 0.329 | 0.363 | 0.343 | 0.377 | 0.324 | 0.365 | 0.345 | 0.372 |
| | 192 | 0.355 | 0.379 | 0.397 | 0.402 | 0.374 | 0.389 | 0.382 | 0.391 | 0.360 | 0.382 | 0.376 | 0.397 | 0.370 | 0.388 | 0.381 | 0.395 | 0.367 | 0.389 | 0.382 | 0.391 |
| | 336 | 0.392 | 0.405 | 0.427 | 0.427 | 0.405 | 0.409 | 0.417 | 0.418 | 0.388 | 0.403 | 0.417 | 0.422 | 0.396 | 0.407 | 0.419 | 0.418 | 0.400 | 0.409 | 0.414 | 0.414 |
| | 720 | 0.448 | 0.441 | 0.493 | 0.463 | 0.466 | 0.444 | 0.501 | 0.461 | 0.456 | 0.438 | 0.481 | 0.458 | 0.465 | 0.447 | 0.487 | 0.457 | 0.460 | 0.445 | 0.473 | 0.450 |
| | Avg | 0.377 | 0.395 | 0.412 | 0.416 | 0.395 | 0.403 | 0.409 | 0.410 | 0.381 | 0.396 | 0.398 | 0.411 | 0.390 | 0.401 | 0.407 | 0.412 | 0.388 | 0.402 | 0.403 | 0.407 |
| ETTm2 | 96 | 0.176 | 0.259 | 0.185 | 0.265 | 0.182 | 0.267 | 0.174 | 0.255 | 0.171 | 0.257 | 0.177 | 0.262 | 0.177 | 0.258 | 0.185 | 0.271 | 0.182 | 0.266 | 0.194 | 0.293 |
| | 192 | 0.241 | 0.302 | 0.256 | 0.310 | 0.247 | 0.308 | 0.249 | 0.304 | 0.237 | 0.300 | 0.247 | 0.307 | 0.241 | 0.302 | 0.254 | 0.314 | 0.250 | 0.311 | 0.283 | 0.360 |
| | 336 | 0.297 | 0.339 | 0.314 | 0.345 | 0.309 | 0.346 | 0.319 | 0.349 | 0.296 | 0.338 | 0.312 | 0.346 | 0.305 | 0.341 | 0.315 | 0.352 | 0.313 | 0.350 | 0.376 | 0.423 |
| | 720 | 0.397 | 0.396 | 0.424 | 0.412 | 0.409 | 0.402 | 0.418 | 0.405 | 0.397 | 0.397 | 0.414 | 0.403 | 0.402 | 0.398 | 0.413 | 0.407 | 0.417 | 0.412 | 0.529 | 0.509 |
| | Avg | 0.278 | 0.324 | 0.294 | 0.333 | 0.287 | 0.331 | 0.290 | 0.328 | 0.275 | 0.323 | 0.287 | 0.330 | 0.281 | 0.324 | 0.292 | 0.336 | 0.291 | 0.335 | 0.346 | 0.396 |
| ETTh1 | 96 | 0.374 | 0.399 | 0.409 | 0.425 | 0.373 | 0.392 | 0.382 | 0.403 | 0.384 | 0.401 | 0.390 | 0.411 | 0.375 | 0.394 | 0.394 | 0.409 | 0.381 | 0.400 | 0.396 | 0.411 |
| | 192 | 0.428 | 0.431 | 0.469 | 0.460 | 0.420 | 0.420 | 0.436 | 0.434 | 0.444 | 0.431 | 0.442 | 0.442 | 0.430 | 0.425 | 0.447 | 0.440 | 0.429 | 0.433 | 0.446 | 0.441 |
| | 336 | 0.465 | 0.457 | 0.507 | 0.478 | 0.459 | 0.441 | 0.492 | 0.455 | 0.490 | 0.452 | 0.480 | 0.468 | 0.478 | 0.446 | 0.490 | 0.464 | 0.475 | 0.460 | 0.490 | 0.468 |
| | 720 | 0.471 | 0.470 | 0.521 | 0.497 | 0.488 | 0.480 | 0.508 | 0.490 | 0.537 | 0.498 | 0.494 | 0.488 | 0.507 | 0.485 | 0.517 | 0.501 | 0.517 | 0.501 | 0.513 | 0.511 |
| | Avg | 0.435 | 0.439 | 0.476 | 0.465 | 0.435 | 0.433 | 0.455 | 0.446 | 0.464 | 0.446 | 0.452 | 0.452 | 0.448 | 0.437 | 0.462 | 0.454 | 0.451 | 0.449 | 0.461 | 0.458 |
| ETTh2 | 96 | 0.295 | 0.347 | 0.331 | 0.372 | 0.293 | 0.345 | 0.312 | 0.358 | 0.297 | 0.343 | 0.328 | 0.371 | 0.290 | 0.339 | 0.300 | 0.350 | 0.301 | 0.351 | 0.348 | 0.401 |
| | 192 | 0.367 | 0.392 | 0.429 | 0.423 | 0.370 | 0.392 | 0.388 | 0.403 | 0.381 | 0.400 | 0.402 | 0.414 | 0.385 | 0.400 | 0.380 | 0.399 | 0.374 | 0.398 | 0.473 | 0.474 |
| | 336 | 0.408 | 0.429 | 0.450 | 0.451 | 0.415 | 0.428 | 0.443 | 0.443 | 0.430 | 0.440 | 0.435 | 0.443 | 0.426 | 0.434 | 0.422 | 0.432 | 0.429 | 0.439 | 0.588 | 0.539 |
| | 720 | 0.420 | 0.438 | 0.459 | 0.466 | 0.425 | 0.443 | 0.455 | 0.459 | 0.466 | 0.465 | 0.417 | 0.441 | 0.448 | 0.451 | 0.429 | 0.447 | 0.443 | 0.461 | 0.829 | 0.656 |
| | Avg | 0.373 | 0.402 | 0.417 | 0.428 | 0.376 | 0.402 | 0.400 | 0.415 | 0.393 | 0.412 | 0.396 | 0.417 | 0.387 | 0.406 | 0.383 | 0.407 | 0.387 | 0.412 | 0.559 | 0.518 |
| Weather | 96 | 0.162 | 0.209 | 0.171 | 0.222 | 0.171 | 0.212 | 0.157 | 0.201 | 0.158 | 0.204 | 0.163 | 0.212 | 0.162 | 0.207 | 0.176 | 0.215 | 0.177 | 0.219 | 0.197 | 0.258 |
| | 192 | 0.208 | 0.251 | 0.234 | 0.273 | 0.219 | 0.254 | 0.244 | 0.273 | 0.205 | 0.248 | 0.212 | 0.254 | 0.221 | 0.257 | 0.226 | 0.258 | 0.222 | 0.258 | 0.237 | 0.296 |
| | 336 | 0.263 | 0.291 | 0.284 | 0.306 | 0.275 | 0.296 | 0.283 | 0.303 | 0.264 | 0.292 | 0.272 | 0.299 | 0.267 | 0.289 | 0.281 | 0.299 | 0.281 | 0.299 | 0.282 | 0.332 |
| | 720 | 0.343 | 0.344 | 0.358 | 0.352 | 0.352 | 0.346 | 0.364 | 0.354 | 0.344 | 0.345 | 0.350 | 0.348 | 0.346 | 0.344 | 0.359 | 0.350 | 0.355 | 0.348 | 0.347 | 0.385 |
| | Avg | 0.244 | 0.274 | 0.262 | 0.288 | 0.254 | 0.277 | 0.262 | 0.283 | 0.243 | 0.272 | 0.249 | 0.278 | 0.249 | 0.275 | 0.260 | 0.281 | 0.259 | 0.281 | 0.266 | 0.318 |
| ECL | 96 | 0.138 | 0.233 | 0.167 | 0.271 | 0.190 | 0.272 | 0.151 | 0.245 | 0.137 | 0.234 | 0.165 | 0.274 | 0.141 | 0.238 | 0.148 | 0.240 | 0.180 | 0.272 | 0.210 | 0.302 |
| | 192 | 0.155 | 0.248 | 0.186 | 0.288 | 0.198 | 0.283 | 0.168 | 0.259 | 0.160 | 0.255 | 0.184 | 0.292 | 0.159 | 0.253 | 0.165 | 0.256 | 0.188 | 0.279 | 0.210 | 0.305 |
| | 336 | 0.172 | 0.266 | 0.203 | 0.304 | 0.217 | 0.303 | 0.184 | 0.268 | 0.173 | 0.270 | 0.195 | 0.302 | 0.178 | 0.273 | 0.179 | 0.271 | 0.204 | 0.295 | 0.223 | 0.319 |
| | 720 | 0.202 | 0.294 | 0.227 | 0.322 | 0.265 | 0.341 | 0.207 | 0.297 | 0.198 | 0.292 | 0.231 | 0.332 | 0.230 | 0.318 | 0.208 | 0.298 | 0.245 | 0.328 | 0.258 | 0.350 |
| | Avg | 0.167 | 0.260 | 0.196 | 0.296 | 0.218 | 0.300 | 0.177 | 0.267 | 0.167 | 0.263 | 0.194 | 0.300 | 0.177 | 0.270 | 0.175 | 0.267 | 0.204 | 0.294 | 0.225 | 0.319 |
| Exchange | 96 | 0.081 | 0.199 | 0.115 | 0.246 | 0.082 | 0.200 | 0.083 | 0.199 | 0.087 | 0.203 | 0.102 | 0.230 | 0.087 | 0.204 | 0.094 | 0.216 | 0.088 | 0.205 | 0.093 | 0.226 |
| | 192 | 0.173 | 0.295 | 0.213 | 0.335 | 0.173 | 0.295 | 0.190 | 0.307 | 0.186 | 0.306 | 0.195 | 0.317 | 0.178 | 0.300 | 0.184 | 0.307 | 0.189 | 0.309 | 0.184 | 0.324 |
| | 336 | 0.320 | 0.407 | 0.367 | 0.440 | 0.324 | 0.411 | 0.401 | 0.458 | 0.334 | 0.417 | 0.359 | 0.436 | 0.340 | 0.420 | 0.336 | 0.422 | 0.327 | 0.415 | 0.328 | 0.435 |
| | 720 | 0.823 | 0.683 | 0.978 | 0.753 | 0.833 | 0.686 | 0.879 | 0.702 | 0.897 | 0.713 | 0.940 | 0.738 | 0.849 | 0.690 | 0.893 | 0.716 | 0.886 | 0.706 | 0.879 | 0.705 |
| | Avg | 0.349 | 0.396 | 0.418 | 0.443 | 0.353 | 0.398 | 0.388 | 0.417 | 0.376 | 0.410 | 0.399 | 0.430 | 0.363 | 0.404 | 0.377 | 0.415 | 0.373 | 0.409 | 0.371 | 0.423 |
| Average | | 0.317 | 0.356 | 0.354 | 0.381 | 0.331 | 0.363 | 0.340 | 0.367 | 0.328 | 0.360 | 0.339 | 0.374 | 0.328 | 0.360 | 0.337 | 0.367 | 0.336 | 0.369 | 0.376 | 0.405 |

Table 9: Full results of short term forecasting task on **pattern segement-based models** and **representative forecasting models**. The prediction lenghs are {6, 48} and results are weighted averaged from several datasets under different sample intervals. **Red**: best, Blue: second best. (* means former, TMixer is TimeMixer)

| Methods Metircs | | NoP (**Ours**) | TimesNet (2023) | Peri-mid* (2024) | iTrans* (2023) | PatchTST (2022) | DLinear (2023) | LightTS (2022) | Pyra* (2022a) | Stationary (2022b) | FED* (2022) | TMixer (2024b) | Re* (2020) |
|---|---|---|---|---|---|---|---|---|---|---|---|---|---|
| Year. | SMAPE | 13.386 | 13.463 | 13.483 | 13.724 | 13.677 | 14.340 | 13.444 | 14.594 | 13.717 | 13.508 | **13.369** | 13.752 |
| | MASE | 3.043 | 3.058 | 3.080 | 3.157 | 3.049 | 3.112 | 3.022 | 3.269 | 3.078 | 3.051 | **3.009** | 3.088 |
| | OWA | 0.792 | 0.797 | 0.800 | 0.817 | 0.802 | 0.830 | **0.792** | 0.858 | 0.807 | 0.797 | **0.787** | 0.809 |
| Quart. | SMAPE | 10.055 | 10.069 | **10.037** | 13.473 | 10.922 | 10.510 | 10.252 | 11.654 | 10.958 | 10.706 | 10.131 | 10.900 |
| | MASE | 1.178 | 1.175 | **1.170** | 1.722 | 1.326 | 1.241 | 1.183 | 1.392 | 1.325 | 1.263 | 1.186 | 1.316 |
| | OWA | 0.886 | 0.886 | **0.882** | 1.240 | 0.979 | 0.930 | 0.897 | 1.037 | 0.981 | 0.947 | 0.893 | 0.975 |
| Month. | SMAPE | **12.713** | 12.760 | 12.795 | 13.674 | 14.200 | 13.382 | 12.798 | 14.963 | 13.917 | 13.925 | 12.762 | 13.949 |
| | MASE | **0.939** | 0.947 | 0.948 | 1.068 | 1.111 | 1.007 | 0.957 | 1.165 | 1.097 | 1.062 | 0.940 | 1.096 |
| | OWA | **0.882** | 0.887 | 0.889 | 0.976 | 1.015 | 0.937 | 0.894 | 1.066 | 0.998 | 0.982 | 0.884 | 0.999 |
| Others. | SMAPE | **4.855** | 4.995 | 4.912 | 5.598 | 5.658 | 5.122 | 5.324 | 5.605 | 6.302 | 4.888 | 5.085 | 6.611 |
| | MASE | 3.256 | 3.346 | 3.260 | 3.957 | 3.626 | 3.608 | 3.410 | 3.966 | 4.064 | **3.244** | 3.403 | 4.492 |
| | OWA | **1.024** | 1.053 | 1.031 | 1.213 | 1.167 | 1.108 | 1.098 | 1.215 | 1.304 | 1.026 | 1.072 | 1.404 |
| Avg. | SMAPE | **11.837** | 11.888 | 11.897 | 13.233 | 12.866 | 12.500 | 11.962 | 13.616 | 12.780 | 12.605 | 11.885 | 12.805 |
| | MASE | **1.596** | 1.607 | 1.607 | 1.850 | 1.734 | 1.678 | 1.609 | 1.843 | 1.756 | 1.677 | 1.598 | 1.777 |
| | OWA | **0.854** | 0.858 | 0.859 | 0.972 | 0.928 | 0.899 | 0.862 | 0.984 | 0.930 | 0.903 | 0.856 | 0.937 |

