# OpenReview forum: "Network of Patterns: Time Series Forecasting with Pattern Passing"
_ICLR.cc/2026/Conference — ICLR 2026 Conference Withdrawn Submission_

### Official Review · Reviewer_Di4M · 2025-10-18

**Soundness:** 2
**Presentation:** 3
**Contribution:** 3
**Rating:** 4
**Confidence:** 4

**Summary:**

This paper introduces NoP, a novel method for time series forecasting that organizes multi-scale pattern segments into a network structure, using Spectrum KL Divergence to measure the similarity. A Pattern Passing mechanism is then employed to aggregate information. The authors claim that NoP overcomes the limitations of traditional chain- and tree-based structures by enabling more flexible and comprehensive interactions between patterns. Extensive experiments on multiple benchmarks show that NoP achieves state-of-the-art performance, validating the effectiveness of it.

**Strengths:**

* **Novelty**

The core idea of organizing pattern segments into a network, as opposed to chain or tree structures, is novel and represents a creative contribution to the field of time series forecasting.

* **Empirical Results**

The NoP demonstrates superior performance over a wide range of baselines, including recent pattern-based and other forecasting methods, across both long- and short-term tasks. The ablation studies further substantiate the importance of each component.

* **Thorough Evaluation**

The authors have conducted a comprehensive set of experiments, including comparisons with different structural variants (chain, tree, network), hyperparameter sensitivity and efficiency analysis.

**Weaknesses:**

* **Insufficient Motivation and Theoretical Justification**

The paper's primary weakness is the lack of a compelling theoretical motivation for why a network structure is necessary and fundamentally more suitable than chain or tree structures. The authors empirically show that network-based method performs better. However, this argument remains largely heuristic. The paper would be significantly strengthened by a more rigorous theoretical discussion or analysis that explains the inherent advantages of the NoP.

* **Limited Analysis of Model Effectiveness**

While the paper shows that the method works, it falls short of deeply analyzing why. For instance, beyond the visualization in Figure 5, a more detailed analysis of the properties of the learned pattern network would be insightful.

* **Lack of Computational Complexity Analysis**

The paper mentions the efficiency of NoP but does not provide a detailed computational complexity analysis.

* **Reproducibility Concerns**

The paper includes a "Reproducibility Statement" (on page 10, and I’m not sure if this violates the rule that "At the time of submission, the main text should be 9 pages or fewer"). However, at the time of review, no code or implementation is provided.

**Questions:**

See Weaknesses.

---

### Official Review · Reviewer_ySaJ · 2025-10-22

**Soundness:** 2
**Presentation:** 2
**Contribution:** 2
**Rating:** 2
**Confidence:** 4

**Summary:**

This paper proposes the Network of Patterns (NoP) for time series forecasting, which flexibly connects all relevant pattern segments to enable comprehensive interactions while employing a Pattern Passing strategy to efficiently propagate information.

**Strengths:**

The paper presents a complete structure and introduces SegmentTST as an auxiliary study, outlining a relatively clear research pathway.

**Weaknesses:**

- In the last sentence of the third paragraph in the Introduction, the reference should be made to Figure 2(b) rather than Figure 2(a).

  On the other hand, the explanation based on *dependency* is not convincing — a low attention score may simply result from high similarity and redundant information between the two elements. This raises concerns about the fundamental motivation of the work.

  Moreover, even if the explanation were valid, the rationale for removing the constraints of chain or tree structures remains weak, since a graph structure can naturally be regarded as a generalization of both. The work also lacks realistic motivations such as temporal dependency or hierarchical periodicity.

  Without solid reasoning, the model design appears to offer limited novelty, as the objective seems not to address a genuine problem but merely to pursue a higher SOTA performance.

- Comments on the writing of Method section:

  1. The subsections are poorly connected, resulting in a strong sense of fragmentation. Readers may find it difficult to understand the purpose of each part when reading, which hinders overall readability.
  2. It is recommended to provide the shape changes of the variables throughout the pipeline to enhance clarity and reproducibility.
  3. The authors should explicitly state that the model is constructed using only the seasonal component and explain the rationale behind this design choice, which would improve both readability and methodological rigor.
  4. For example, the opening sentence of Section 3.1 lacks fluency and could be revised to improve readability and clarity.

- Regarding the virtual pattern node proposed in Section 3.3:

  The authors arbitrarily introduce an additional node, and the ablation study demonstrates its effectiveness. From another perspective, this suggests that the graph construction may be incomplete. The authors do not clarify how the virtual node connects with other nodes. If the connections are bidirectional, it effectively introduces additional edges between existing nodes, which could be seen as evidence of imperfect graph construction. If the connections are unidirectional (as illustrated in Figure 3), the aggregation appears to be meaningless. The authors should provide evidence that the graph construction is sufficiently complete—for example, by considering alternative graph construction methods and corresponding experiments with and without the virtual pattern node.

- Lack of experimental convincibility: The manuscript lacks comparisons with strong baselines such as xPatch. Moreover, given the rapid development of large-scale time series models, it is necessary to include comparisons with models like Moirai，Time-MoE. In scenarios with sufficiently long input sequences, the full-shot performance of the proposed model should surpass the zero-shot performance of these models to provide convincing evidence of its effectiveness.

  Additionally, I recommend that the authors supplement their experiments with results on other datasets, such as Monash, since the current benchmark is somewhat controversial (https://cbergmeir.com/talks/neurips2024/) and its performance has reached a high level of saturation (https://arxiv.org/abs/2510.02729). Including datasets like Traffic could further enhance the credibility of the experimental evaluation.

**Questions:**

The main points are detailed in the Weaknesses section. The authors should reiterate the core motivation of their work, provide additional experiments to demonstrate the effectiveness of the virtual pattern node design, furnish further evidence to validate the overall model, and improve the clarity and completeness of the writing.

---

### Official Review · Reviewer_dn8b · 2025-10-30

**Soundness:** 2
**Presentation:** 2
**Contribution:** 2
**Rating:** 2
**Confidence:** 4

**Summary:**

This paper proposes a novel time series forecasting framework, Network of Patterns (NoP). It breaks through the limitations of traditional chain and tree-based pattern aggregation by measuring pattern similarity in the frequency domain (Spectrum KL Divergence) and organizing multi-scale time segments into a network structure. Furthermore, the paper designs a Pattern Passing mechanism, which enables flexible transmission and fusion of cross-scale information between network nodes, thereby achieving efficient modeling of complex cycles and multi-scale dependencies.

**Strengths:**

It is proposed to organize the multi-scale patterns of time series in the form of a "Network-of-Patterns (NoP)", rather than the traditional chain or tree structures. And the logical structure of Introduction, Related Work, Methodology, Experiments, Ablation Studies, and Appendix is rigorous. The ablation experiments (w/o FFT, w/o Network, w/o PPB, w/o Virtual Pattern) fully verify the effectiveness of each module.

**Weaknesses:**

How does the time complexity of SKL computation and network construction change with sequence length or the number of nodes? Can it be extended to ultra-long sequences (i.e., 336, 720)?
In long term forecasting task, according to Table 8 in TimeFilter [1] and Table 8 in your manuscript, under the same settings, the performance of the proposed NoP model is obviously weaker than that of TimeFilter.
In short term forecasting task, Table 4 in TimeMixer [2] shows that the model significantly outperforms TimesNet [3]. Why does TimeMixer perform worse than TimesNet in your manuscript instead? Besides, according to Table 4 in TimeMixer and Table 9 in your manuscript, under the same settings, the performance of the proposed NoP model is obviously weaker than that of TimeMixer and TimesNet. Is there any down-tuning of the baseline here?


[1] Hu Y, Zhang G, Liu P, et al. TimeFilter: Patch-specific spatial-temporal graph filtration for time series forecasting[J]. arXiv preprint arXiv:2501.13041, 2025.
[2] Wang S, Wu H, Shi X, et al. Timemixer: Decomposable multiscale mixing for time series forecasting[J]. arXiv preprint arXiv:2405.14616, 2024.
[3] Wu H, Hu T, Liu Y, et al. Timesnet: Temporal 2d-variation modeling for general time series analysis[J]. arXiv preprint arXiv:2210.02186, 2022.

**Questions:**

As in Weaknesses.

---

### Official Review · Reviewer_bwbi · 2025-11-01

**Soundness:** 2
**Presentation:** 3
**Contribution:** 2
**Rating:** 2
**Confidence:** 5

**Summary:**

This paper proposes a time series forecasting method called NoP, which decomposes the sequence into multi-scale pattern segments, constructs a network structure among the segments using frequency-domain metrics, and aggregates information through a pattern propagation mechanism.

**Strengths:**

S1: All the modules used in this paper are relatively mature, and the overall model design is fairly reasonable.

S2: The description in the paper is clear.

**Weaknesses:**

W1: Although the authors pay attention to some pattern segment-based methods, they overlook several important baseline models based on periodic modeling, such as the linear-based CycleNet[1] and the RNN-based PGN[2]. Moreover, these methods are not included in the experimental comparisons, which limits the comprehensiveness of the model performance evaluation.

[1] CycleNet: Enhancing Time Series Forecasting through Modeling Periodic Patterns. In The Thirty-eighth Annual Conference on Neural Information Processing Systems.

[2] PGN: The RNN's New Successor is Effective for Long-Range Time Series Forecasting. In The Thirty-eighth Annual Conference on Neural Information Processing Systems.

W2: The experimental section of the paper has significant flaws due to the lack of clear description of the hyperparameter search space. It remains unclear whether the authors conducted a standard hyperparameter tuning procedure for all baseline methods (i.e., optimizing hyperparameters on the validation set and reporting final results on the test set). To ensure reproducibility and fairness, the authors should provide the following:

(a) If hyperparameter search was conducted, please provide the complete search space and the final selected parameter values for each task across different datasets;

(b) If no systematic hyperparameter tuning was performed, the credibility of the current experimental results is questionable. Since different hardware platforms can affect model performance, to ensure a rigorous comparison, all baseline methods should be rerun on the authors’ unified experimental platform, tuned using the same broad hyperparameter search space, and the best parameters selected on the validation set should then be evaluated on the test set. Without these supplementary experiments, the reliability of the current conclusions cannot be established.

W3: The efficiency experiments presented by the authors are also clearly insufficient. Although the paper claims that NoP achieves lower MSE with fewer parameters, it is unclear whether this conclusion holds across all tasks. Additionally, it is not stated whether key structural hyperparameters of all baseline models (e.g., hidden dimensions, number of layers) were unified during the efficiency comparison. To perform a fair efficiency analysis, the impact of differences in model scale must be controlled, which differs from the logic of experiments aimed solely at validating model performance. Otherwise, deliberately choosing extremely small parameter configurations for certain tasks to claim “high efficiency” would seriously compromise the rigor of the experiments and the fairness of the conclusions.

**Questions:**

See Weaknesses.

---

### Note · Authors · 2025-11-24

I have read and agree with the venue's withdrawal policy on behalf of myself and my co-authors.